# Prevalence of respiratory viruses using polymerase chain reaction in children with wheezing, a systematic review and meta– analysis

Cyprien Kengne–Nde[1], Sebastien Kenmoe[2], Abdou Fatawou Modiyinji[2,3], Richard Njouom[2]*

1 National AIDS Control Committee, Epidemiological Surveillance, Evaluation and Research Unit, Yaounde, Cameroon, 2 Department of Virology, Centre Pasteur of Cameroon, Yaoundé, Cameroon, 3 Faculty of Sciences, Department of Animals Biology and Physiology, University of Yaoundé I, Yaoundé, Cameroon

* njouom@pasteur–yaounde.org, njouom@yahoo.com

## Abstract

### Introduction

Wheezing is a major problem in children, and respiratory viruses are often believed to be the causative agent. While molecular detection tools enable identification of respiratory viruses in wheezing children, it remains unclear if and how these viruses are associated with wheezing. The objective of this systematic review is to clarify the prevalence of different respiratory viruses in children with wheezing.

### Methods

We performed an electronic in Pubmed and Global Index Medicus on 01 July 2019 and manual search. We performed search of studies that have detected common respiratory viruses in children ≤18 years with wheezing. We included only studies using polymerase chain reaction (PCR) assays. Study data were extracted and the quality of articles assessed. We conducted sensitivity, subgroup, publication bias, and heterogeneity analyses using a random effects model.

### Results

The systematic review included 33 studies. Rhinovirus, with a prevalence of 35.6% (95% CI 24.6–47.3, $I^2$ 98.4%), and respiratory syncytial virus, at 31.0% (95% CI 19.9–43.3, $I^2$ 96.4%), were the most common viruses detected. The prevalence of other respiratory viruses was as follows: human bocavirus 8.1% (95% CI 5.3–11.3, $I^2$ 84.6%), human adenovirus 7.7% (95% CI 2.6–15.0, $I^2$ 91.0%), influenza virus6.5% (95% CI 2.2–12.6, $I^2$ 92.4%), human metapneumovirus5.8% (95% CI 3.4–8.8, $I^2$ 89.0%), enterovirus 4.3% (95% CI 0.1– 12.9, $I^2$ 96.2%), human parainfluenza virus 3.8% (95% CI 1.5–6.9, $I^2$ 79.1%), and human coronavirus 2.2% (95% CI 0.6–4.4, $I^2$ 79.4%).

**Data Availability Statement:** All relevant data are within the manuscript and its Supporting Information files.

**Funding:** The author(s) received no specific funding for this work.

**Competing interests:** The authors have declared that no competing interests exist.

## Conclusions

Our results suggest that rhinovirus and respiratory syncytial virus may contribute to the etiology of wheezing in children. While the clinical implications of molecular detection of respiratory viruses remains an interesting question, this study helps to illuminate the potential of role respiratory viruses in pediatric wheezing.

## Review registration

PROSPERO, CRD42018115128.

## Introduction

Wheezing is a common health challenge in early childhood.[1] More than one–third of children aged < 2 years experience at least one episode of wheezing, and one–fifth experience recurrent wheezing [1]. Pediatric wheezing can sometimes be severe enough to justify hospitalization, admission to intensive care units, and mechanical ventilation [2,3]. Studies show that children with a history of lower respiratory tract infections, primarily bronchiolitis, have an increased risk of developing transient wheezing up to age 13 [4,5]. Wheezing may additionally be related to the development of asthma in adulthood [4,6].

Wheezing is believed to be induced by viral infection [7,8]. Common viruses associated with wheezing include rhinovirus (RV), human respiratory syncytial virus (HRSV), human metapneumovirus (HMPV), human parainfluenza virus (HPIV), enterovirus (EV), human adenovirus (HAdV), human bocavirus (HBoV), human coronavirus (HCoV), and influenza virus [9,10].

Traditional respiratory viral diagnostic methods, such as cell culture and serology, have limitations. For example, RV is difficult to isolate by cell culture [11,12], and the large number of RV serotypes poses a major challenge for serological assays [13]. Compared to molecular detection, traditional diagnostic assays have a lower sensitivity for all common respiratory viruses [14,15]. These traditional diagnostics, however, have commonly been used to document respiratory viral prevalence, particularly in early literature [10]. Consequently, initial studies may have over or understated the role of certain respiratory viruses in pediatric wheezing. Molecular detection has revealed that RV may play a substantive role in the clinical manifestation of respiratory disease than original thought [16,17]. Additionally, molecular detection helped to identify the presence of new viruses such as HMPV and HBoV in children with respiratory signs [18,19].

Thus, the prevalence of common respiratory viruses in wheezing still remains unclear. Assessing the impact of the upcoming HRSV vaccination will require reliable data on prevalence [20]. Accurate viral prevalence data could contribute to optimizing and controlling antibiotic use, and providing better guidance for therapeutic decision making. The aim of this systematic review is to synthesize the prevalence of respiratory viruses in wheezing children, from publications that use molecular tools.

## Methods

### Study design

The study was performed according to the preferred reporting items for systematic reviews and meta–analyses(PRISMA) guidelines (S1 Table) [21]. Ethical approval was not required because the study does not involve the inclusion of humans and/or animals. The present study

was registered in the PROSPERO database (PROSPERO: CRD42018115128;https://www.crd.york.ac.uk/prospero/display_record.php?RecordID=115128).

## Inclusion criteria

We included clinical trial, cohort, case–control, and cross–sectional studies that reported the prevalence of respiratory viruses in children ≤18 years presenting with wheezing. The definitions of wheezing were adapted as described by the authors of the primary studies. In the case of duplicate studies, where the same population was recruited and examined during the same period, only the most recent or complete study was included.

## Exclusion criteria

Reviews, letters to the editor, studies with interrupted study periods, and studies of patients with underlying medical conditions such as bronchopulmonary dysplasia, cystic fibrosis and bronchiolitis obliterans were excluded. We also excluded studies that used non–PCR based methods for viral detection such as culture, time–resolved fluoroimmunoassay, enzyme immunoassay or immunofluorescence. Included articles with multiple follow up time were used only once for each virus.

## Search strategy

Relevant studies were identified through research conducted in Pubmed and Global Index Medicus. We used search terms associated with the common respiratory viruses including RV, EV, HRSV, HMPV, HPIV, HCoV, HAdV, HBoV, and influenza and with the clinical signs of wheezing. The search string used in Pubmed and Global Index Medicus is illustrated in S2 Table. The search was carried out without any language restrictions and considered publication dates until 01 July 2019. We used Google Translate for articles written in languages other than English and French. Study reference sections and relevant review articles were used to identify additional articles.

## Study selection

Two investigators (SK and AFM) independently selected eligible studies based on the titles and abstracts from the list of references on the Rayyan website, a free web–based application used to assist authors of systematic reviews (https://rayyan.qcri.org/welcome) [22]. The selection process was summarized in a PRISMA flow chart [21].

## Data extraction

The complete versions of the selected articles were downloaded and reviewed by two authors (AFM and SK) of the study. The data were extracted using a pre–designed abstraction form, specifically we collected: study design, study country, World Health Organization (WHO) region, sampling method, mean age of study participants, percentage of male patients, total number of patients tested, and total number of positive samples for each virus.

## Appraisal of methodological quality and risk of bias

Two authors (AFM and SK) independently assessed the quality of each study using the Hoy et al assessment scale [23]. This scale has 10 dichotomous questions that assess the internal and external validity of a study (S3 Table). According to these questions, articles were classified as low, moderate, or high risk of bias.

## Data synthesis and analysis

Disagreements in the selection of studies, data extraction and evaluation of study quality were resolved by discussion and consensus among the authors. Inter rater agreement for study selection was calculated using the kappa statistic Kappa values [−1–0], [0–0.2], [>0.2–0.4], [>0.4–0.6], [>0.6–0.8] and [>0.8–1] represented an extremely weak, very weak, weak, medium, satisfactory and excellent inter–rater agreement respectively [24]. Data were analyzed using the 'meta' package of the statistical software R (version 3.5.1) [25,26]. Unadjusted prevalence has been recalculated based on the information of crude numerators and denominators provided by individual studies. Prevalence was reported with a 95% confidence interval (CI) and a 95% prediction interval (PI). The variance of the study specific prevalence was stabilized with the Freeman–Tukey dual arcsine transformation before pooling the data within a random–effects meta–analysis model [27]. Egger's test served to assess the presence of publication bias [28]. A p–value of <0.10 for the Egger test was considered statistically significant. Heterogeneity was evaluated by the $\chi^2$ test on Cochrane's Q statistic [29], which was quantified by H and $I^2$ values. The $I^2$ statistic estimates the percentage of total variation across studies due to true between–study differences rather than chance. In general, $I^2$ values >70% indicate the presence of substantial heterogeneity [30]. Subgroup analyses were performed for the following subgroups: WHO region, mean age group (0–2 years, 0–5 years, and 0–18 years), and a p–value <0.05 was considered statistically significant. The effect of variables that could explain the heterogeneity in the included studies was examined by a univariate and multivariate metaregression model. To assess the influence of design and quality of included studies an overall estimated sensitivity analysis was performed using only cross-sectional studies and studies with low risk of bias.

## Results

### Study selection and characteristics

Initially, we identified 1426 publication abstracts and removed 28 duplicates (Fig 1). After screening, we excluded 1247 that we found to be irrelevant, most of which were describing patients with asthma, bronchiolitis, atopy, chronic obstructive pulmonary disease, pneumonia or underlying medical conditions; studies on therapy, vaccination, prophylaxis, immune response or animals; and reviews. We assessed the texts of the remaining 151 papers for eligibility, of which 118were excluded most often because respiratory viruses were not investigated in the study (25;21.2%), wheezing patients were not recruited (47;39.8%), or it was an ineligible study type (15;12.7%)(S4 Table). Study selection inter rater agreement was κ = 0.87, representing excellent interrater agreement. Thirty–three full texts were retained for the review and included in the meta–analysis [31–63].

### Characteristics of included studies

Study participants were recruited between January 1992 and November 2014 (Table 1). According to the Shapiro-Wilk test, the age distribution did not follow a normal distribution (p <0.001) and therefore we expressed the age as the median and interquartile range. The cumulative number of samples tested from study participants was 18,365; however, the number of participants tested for each virus was variable. The percentage of males in each study ranged from 50–75%. The majority of participants recruited in included studies (20, 60.6%) were aged <5 years. The studies were published between 2002 and 2018. Most included studies were performed by consecutive sampling (25; 75.8%), had a prospective recruitment (29; 87.9%), had a moderate risk of bias (19; 57.6%), and tested specimens from the nasopharynx (29; 87.9%). Individual characteristics of included studies are presented in S5 Table.

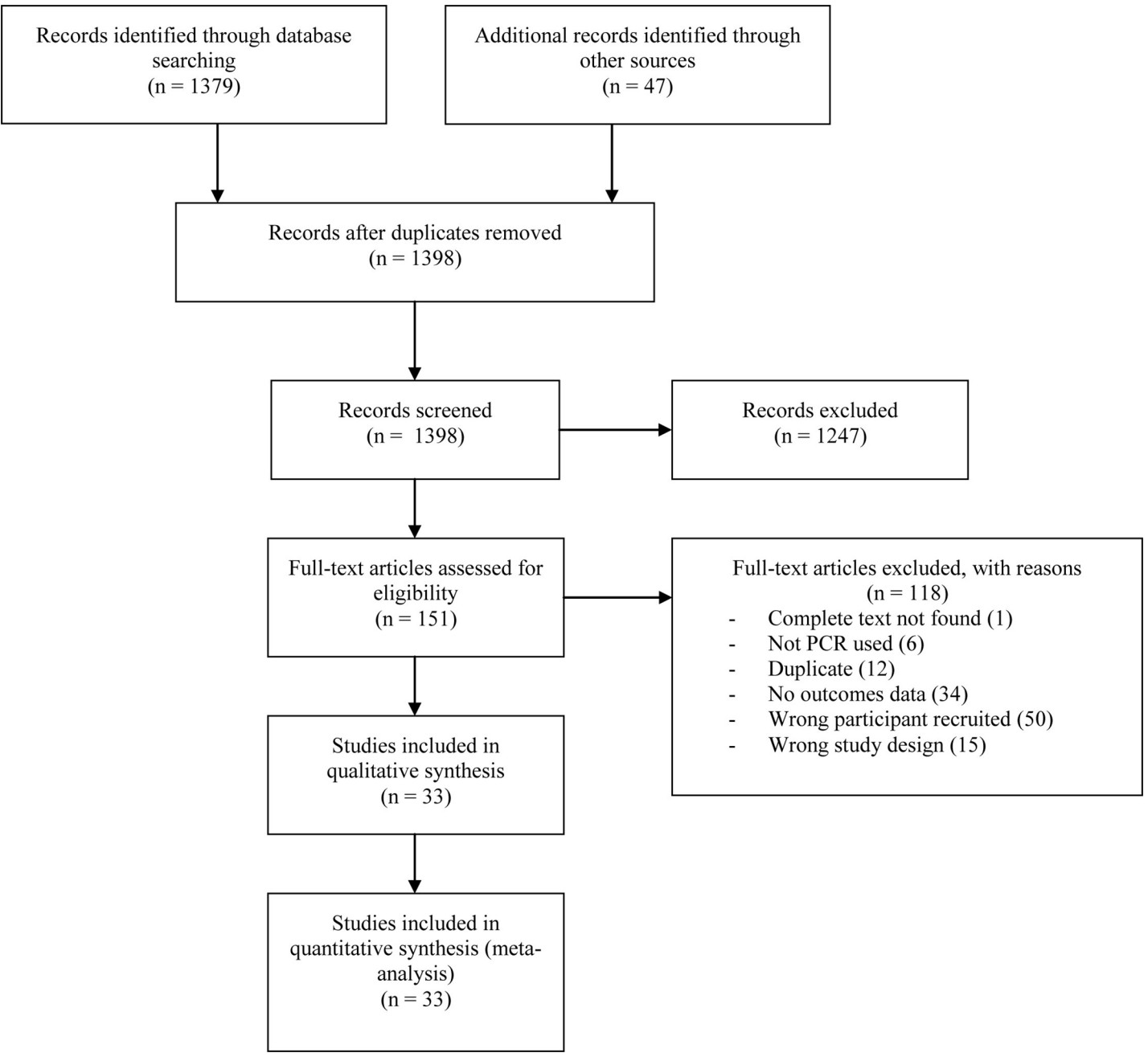

**Fig 1. PRISMA flow chart of literature search and selection process.**

## Prevalence of viral infections among children with wheezing

The prevalence of viruses detected with molecular assays in children with wheezing was: RV 35.6% (95% CI 24.6–47.3,$I^2$ 98.4%), HRSV 31.0% (95% CI 19.9–43.3,$I^2$ 96.4%), HBoV 8.1% (95% CI 5.3–11.3, $I^2$ 84.6%), HAdV 7.7% (95% CI 2.6–15.0, $I^2$ 91.0%),influenza virus 6.5% (95% CI 2.2–12.6,$I^2$ 92.4%),HMPV 5.8% (95% CI 3.4–8.8, $I^2$ 89.0%), EV 4.3% (95% CI 0.1–12.9, $I^2$96.2%), HPIV 3.8% (95% CI 1.5–6.9, $I^2$ 79.1%), and HCoV 2.2% (95% CI 0.6–4.4, $I^2$ 79.4%) (Fig 2 and S1–S9 Figs). When sensitivity analyses were conducted for risk of bias and

**Table 1. Sociodemographic and clinical characteristics of included studies.**

| Characteristics | N = 33 |
|---|---|
| %Male. range | 50.1–75 |
| Age (years) | Mean: 1.5 (IQR: [1.0–3.2]) |
| Age range | |
| • < 2 years | 10 (30%) |
| • < 5 years | 10 (30%) |
| • 0–18 years | 9 (27%) |
| • Unclear/Not described | 5 (15%) |
| Period of inclusion of participants. range | Jan/1992-Nov/2014 |
| Year of publication. range | 2002–2018 |
| Study design | |
| • Case control | 1 (3%) |
| • Clinical trial | 7 (21%) |
| • Cohort | 9 (27%) |
| • Cross-sectional | 16 (48%) |
| Sampling | |
| • Consecutive | 25 (75%) |
| • Random | 8 (24%) |
| Timing of data collection | |
| • Prospective | 29 (87%) |
| • Retrospective | 4 (12%) |
| Study bias | |
| • Low risk | 14 (42%) |
| • Moderate risk | 19 (57%) |
| WHO region | |
| • Africa | 2 (6%) |
| • America | 3 (9%) |
| • Eastern Mediterranean | 1 (3%) |
| • Europe | 15 (45%) |
| • South-East Asia | 2 (6%) |
| • Western Pacific | 10 (30%) |
| Clinical presentation | |
| • Acute wheezing | 30 (91%) |
| • Recurrent wheezing | 3 (9%) |
| Sample type | |
| • Nasal samples | 11 (33%) |
| • Oropharyngeal samples | 4 (13%) |
| • Nasopharyngeal samples | 29 (88%) |
| • Tracheal samples | 2 (6%) |

study design, they did not differ from the overall analysis (Table 2). Considerable heterogeneity was present in overall prevalence and sensitivity analysis for all viruses. Publication bias was present for studies that examined HMPV, HPIV, and HCoV (S10–S18 Figs).

## Subgroup analysis and metaregression

All subgroup analyses are presented in S6 Table. There were significant differences between age group and respiratory virus prevalence for RV (p = 0.024), HAdV (p< 0.001), influenza (p = 0.001) and EV (p = 0.012). Rhinovirus (> 2 year), HAdV (>2 years) and EV (> 2 years)

| Study | Total | | Prevalence (%) | 95% CI |
|---|---|---|---|---|
| **Rhinovirus (24 Studies)** | | | | |
| **Random effect meta–analysis** | **4394** | | 35.62 | [24.67; 47.38] |
| Heterogeneity: $I^2$ = 98.4% [98.1%; 98.6%], $\tau^2$ = 0.0846, $p < 0.0001$ | | | | |
| **HRSV (12 Studies)** | | | | |
| **Random effect meta–analysis** | **1810** | | 31.07 | [19.99; 43.34] |
| Heterogeneity: $I^2$ = 96.4% [95.0%; 97.4%], $\tau^2$ = 0.0471, $p < 0.0001$ | | | | |
| **HBoV (11 Studies)** | | | | |
| **Random effect meta–analysis** | **2592** | | 8.11 | [ 5.35; 11.35] |
| Heterogeneity: $I^2$ = 84.6% [74.0%; 90.8%], $\tau^2$ = 0.0063, $p < 0.0001$ | | | | |
| **HAdV (8 Studies)** | | | | |
| **Random effect meta–analysis** | **1084** | | 7.79 | [ 2.64; 15.06] |
| Heterogeneity: $I^2$ = 91% [84.6%; 94.7%], $\tau^2$ = 0.0227, $p < 0.0001$ | | | | |
| **Influenza (9 Studies)** | | | | |
| **Random effect meta–analysis** | **1380** | | 6.53 | [ 2.23; 12.64] |
| Heterogeneity: $I^2$ = 92.4% [87.8%; 95.3%], $\tau^2$ = 0.0221, $p < 0.0001$ | | | | |
| **HMPV (15 Studies)** | | | | |
| **Random effect meta–analysis** | **3093** | | 5.88 | [ 3.44; 8.87] |
| Heterogeneity: $I^2$ = 89% [83.6%; 92.7%], $\tau^2$ = 0.0104, $p < 0.0001$ | | | | |
| **Enterovirus (6 Studies)** | | | | |
| **Random effect meta–analysis** | **1222** | | 4.31 | [ 0.10; 12.95] |
| Heterogeneity: $I^2$ = 96.2% [93.8%; 97.6%], $\tau^2$ = 0.0355, $p < 0.0001$ | | | | |
| **HPIV (9 Studies)** | | | | |
| **Random effect meta–analysis** | **1280** | | 3.85 | [ 1.57; 6.94] |
| Heterogeneity: $I^2$ = 79.1% [60.8%; 88.9%], $\tau^2$ = 0.0076, $p < 0.0001$ | | | | |
| **HCoV (8 Studies)** | | | | |
| **Random effect meta–analysis** | **1510** | | 2.22 | [ 0.66; 4.49] |
| Heterogeneity: $I^2$ = 79.4% [59.9%; 89.5%], $\tau^2$ = 0.0056, $p < 0.0001$ | | | | |
| **Overall random effect meta–analysis** | **18365** | | 13.61 | [10.23; 17.38] |
| Residual heterogeneity: $I^2$ = 95.9% [95.4%; 96.4%], $p = 0$ | | | | |

0  20  40  60  80

**Fig 2. Global prevalence of respiratory viruses among children aged < 18 years with wheezing, January 1992-November 2014.**

were associated with older ages while influenza (< 1 year) was associated with younger ages. RV (p< 0.001), influenza (p = 0.016), HMPV (p< 0.001), and EV (p = 0.042) prevalence varied significantly according to WHO region. The prevalence of viruses was significantly different

**Table 2. Overall meta-analysis for global prevalence of respiratory viral infections in children with wheezing.**

| | Prevalence. % (95%CI) | 95% Prediction interval | N Studies | N Participants | H (95%CI) | I² (95%CI) | P heterogeneity | P Egger test |
|---|---|---|---|---|---|---|---|---|
| **Rhinovirus** | | | | | | | | |
| Overall | 35.6 [24.6–47.3] | [0.0–90.9] | 24 | 4394 | 7.8 [7.1–8.5] | 98.4 [98.1–98.6] | < 0.001 | 0.632 |
| Low risk | 30.3 [14.0–49.7] | [0.0–96.2] | 11 | 1974 | 9.6 [7.6–9.7] | 98.7 [98.3–99.0] | < 0.001 | 0.932 |
| Cross sectional studies | 30.0 [16.5–45.5] | [0.0–86.7] | 10 | 1659 | 6.3 [5.4–7.4] | 97.5 [96.6–98.2] | < 0.001 | 0.882 |
| **HRSV** | | | | | | | | |
| Overall | 31.0 [19.9–43.3] | [0.3–79.5] | 12 | 1810 | 5.2 [4.4–6.1] | 96.4 [95.0–97.4] | < 0.001 | 0.481 |
| Low risk | 29.3 [15.3–45.6] | [0.0–86.7] | 8 | 1395 | 6.0 [4.9–7.2] | 97.2 [96.0–98.1] | < 0.001 | 0.552 |
| Cross sectional studies | 20.6 [9.6–34.3] | [0.0–73.6] | 6 | 931 | 4.0 [3.1–5.4] | 94.0 [89.6–96.6] | < 0.001 | 0.996 |
| **HBoV** | | | | | | | | |
| Overall | 8.1 [5.3–11.3] | [0.7–21.4] | 11 | 2592 | 2.5 [1.9–3.3] | 84.6 [74.0–90.8] | < 0.001 | 0.677 |
| Low risk | 6.8 [1.6–14.8] | [0.0–59.0] | 4 | 953 | 3.5 [2.4–5.2] | 92.1 [83.0–96.3] | < 0.001 | 0.799 |
| Cross sectional studies | 7.4 [4.6–10.8] | [0.6–19.8] | 8 | 1539 | 2.0 [1.4–2.9] | 76.3 [52.6–88.1] | < 0.001 | 0.248 |
| **HAdV** | | | | | | | | |
| Overall | 7.7 [2.6–15.0] | [0.0–40.1] | 8 | 1084 | 3.3 [2.5–4.3] | 91.0 [84.6–94.7] | < 0.001 | 0.734 |
| Low risk | 11.6 [3.6–23.1] | [0.0–63.3] | 5 | 822 | 3.4 [2.4–4.9] | 91.8 [83.9–95.8] | < 0.001 | 0.565 |
| Cross sectional studies | 9.9 [2.4–21.0] | [0.0–58.9] | 6 | 888 | 3.8 [2.8–5.1] | 93.2 [87.9–96.2] | < 0.001 | 0.672 |
| **Influenza** | | | | | | | | |
| Overall | 6.5 [2.2–12.6] | [0.0–35.4] | 9 | 1380 | 3.6 [2.8–4.6] | 92.4 [87.8–95.3] | < 0.001 | 0.142 |
| Low risk | 8.3 [1.6–19.0] | [0.0–61.4] | 5 | 965 | 4.3 [3.2–5.8] | 94.7 [90.4–97.1] | < 0.001 | 0.184 |
| Cross sectional studies | 6.4 [0.6–16.6] | [0.0–59.0] | 5 | 861 | 3.9 [2.9–5.4] | 93.7 [88.2–96.6] | < 0.001 | 0.226 |
| **HMPV** | | | | | | | | |
| Overall | 5.8 [3.4–8.8] | [0.0–21.2] | 15 | 3093 | 3.0 [2.4–3.6] | 89.0 [83.6–92.7] | < 0.001 | 0.001 |
| Low risk | 4.7 [2.4–7.6] | [0.0–16.2] | 5 | 1086 | 1.7 [1.0–2.7] | 65.7 [10.4–86.9] | 0.020 | 0.071 |
| Cross sectional studies | 7.7 [4.2–11.9] | [0.0–25.7] | 9 | 1612 | 2.7 [2.0–3.5] | 86.4 [76.2–92.2] | < 0.001 | 0.049 |
| **Enterovirus** | | | | | | | | |
| Overall | 4.3 [0.1–12.9] | [0.0–49.9] | 6 | 1222 | 5.1 [4.0–6.5] | 96.2 [93.8–97.6] | < 0.001 | 0.808 |
| Low risk | 5.8 [0.2–16.8] | [0.0–64.7] | 5 | 1107 | 5.4 [4.1–7.0] | 96.6 [94.3–98.0] | < 0.001 | 0.658 |
| Cross sectional studies | 0.4 [0.0–1.2] | NA | 2 | 654 | 1.0 | 0.0 | 0.380 | NA |
| **HPIV** | | | | | | | | |
| Overall | 3.8 [1.5–6.9] | [0.0–17.0] | 9 | 1280 | 2.1 [1.6–3.0] | 79.1 [60.8–88.9] | < 0.001 | 0.081 |

*(Continued)*

**Table 2.** (Continued)

|  | Prevalence. % (95%CI) | 95% Prediction interval | N Studies | N Participants | H (95%CI) | I² (95%CI) | P heterogeneity | P Egger test |
|---|---|---|---|---|---|---|---|---|
| Low risk | 5.5 [1.2–12.2] | [0.0–39.4] | 5 | 865 | 2.9 [2.0–4.2] | 88.2 [75.0–94.4] | < 0.001 | 0.125 |
| Cross sectional studies | 3.8 [0.9–8.3] | [0.0–24.1] | 6 | 931 | 2.3 [1.6–3.4] | 82.5 [63.0–91.7] | < 0.001 | 0.143 |
| **HCoV** |  |  |  |  |  |  |  |  |
| Overall | 2.2 [0.6–4.4] | [0.0–12.1] | 8 | 1510 | 2.2 [1.5–3.0] | 79.4 [59.9–89.5] | < 0.001 | 0.025 |
| Low risk | 1.6 [0.0–4.4] | [0.0–17.2] | 5 | 1107 | 2.4 [1.6–3.6] | 83.2 [61.8–92.6] | < 0.001 | 0.179 |
| Cross sectional studies | 1.7 [0.0–5.7] | [0.0–98.0] | 3 | 869 | 2.7 [1.6–4.6] | 87.0 [62.9–95.4] | < 0.001 | 0.128 |

CI: Confidence interval; RV: Rhinovirus; HCoV: Human Coronavirus; HPIV: Human Parainfluenzavirus; HMPV: Human Metapneumovirus; HRSV: Human Respiratory Syncytial Virus; HAdV: Human Adenovirus; HBoV: Human Bocavirus; EV: Enterovirus; NA: Not applicable.

according to the detection assay used for RV (<0.001), HBoV (p = 0.001) and HCoV (p = 0.007). Compared to conventional RT-PCR, real-time RT-PCR was associated with higher prevalences for HCoV while conventional RT-PCR was associated with higher prevalences for RV and HBoV. There were no differences for the remaining subgroup analyses. Substantial heterogeneity was present most subgroup analyses. The multivariate metaregression model (S7 Table) indicates that the detection technique contributed to the heterogeneity of the results for the prevalence of RV, HRSV, Influenza and HBoV. The age of the participants contributed to the heterogeneity of the results for the prevalence of HAdV, Influenza and HCoV. In agreement with the Egger test and visual inspection of the funnel plot, there was evidence of publication bias for some age subgroups and WHO region for RV, HRSV, HBoV, HMPV and HCoV (S6 Table and S10–S18 Figs).

## Discussion

This study is the first systematic review of respiratory viruses in children with wheezing for nearly 30 years. The findings emphasize the strong association between the detection of RV and HRSV in wheezing children. Deoxyribonucleic acid viruses, HBoV and HAdV, were the second most commonly detected viruses. This study also showed a preponderance of RV, HAdV, and EV in older children and influenza in younger children.

Contrary to common knowledge that RV is associated with asymptomatic infection and mild illness, RV could have an important role in the clinical presentation of pediatric wheezing. As previously reported by the narrative reviews on this topic [7,10,11,64]. RV and HRSV were the most common viruses found in patients with wheezing. Although RVs are not sensitive to all cell lines, and earlier research studies lacked the molecular tools available today, the quantitative review analysis done by Pattemore et al in 1996 also recognized the role of RV in developing wheezing [11]. The hypothesis of the importance of RV in respiratory infections is well supported by two recent systematic reviews [65,66]. One has shown that RV was the predominant virus in asthma which is usually accompanied by wheezing [65]. The second review recently demonstrated that RV infections in the first 3 years of life were significantly associated with a high risk of subsequent wheezing and asthma at pre–school age [66]. In contrast to the Pattemore et al. study, a low prevalence of HPIV has been observed in this review. This could be explained by the inclusion of antibody–detecting studies and the wide range of clinical

definitions in the Pattemore et al [11]. The detection of antibodies in the studies represents both acute and past infections and thus reports the highest prevalence [3,67].

Many studies have shown that HRSV is a major causative agent of wheezing in children under 2 years of age [9,68–70]. Although not significant in our study, the prevalence of HRSV was inversely proportional to child age.

These findings must be considered within the context of study limitations, including a small number of studies that met inclusion criteria, few studies that described wheezing type, and a limited worldwide geographic representation. Wheezing is considered bronchiolitis in children <1 year of age [71]. Almost all of the studies included in this review have children <1 year of age that we cannot exclude and could therefore be the source of additional heterogeneity. The studies included in this systematic review reported the prevalence of EV and RV individually. It is, however, known that EV and RV cross react in real-time PCR assays and this should be taken into account while interpreting the study results [72,73]. Seasonality and study duration could also drive the variability in the prevalence of respiratory viruses. This study, however, did not take these into account because the prevalence data were not always reported by these parameters. Molecular detection alone cannot implicate the etiology of clinical signs, as asymptomatic carriage of some respiratory viruses is possible. For example, the causal role of HBoV in acute respiratory infections, which has been reported in asymptomatic infections and mostly reported in codetection, has been widely controversial to date [74]. Additionally, combinations of viruses detected, or co–detections were beyond the scope of this work.

The present study has multiple strengths, particularly the fact that many respiratory viruses were considered in the analysis and that only studies with molecular viral detection were included, a gold standard and common diagnostic tool for respiratory viruses. In addition, the methodological approach strengthened this analysis, with the inclusion of analyses for explanation of sources of heterogeneity and publication bias. We used a comprehensive search strategy and two independent authors were involved in all stages.

RV and HRSV were the predominant viruses most commonly detected in children with wheezing. HRSV was largely present in children ≤ 2 years. This systematic review also highlights the important role played by newly described viral agents, HBoV and HMPV, in wheezing.

The completion and marketing of an HRSV vaccine as well as the development of antivirals against respiratory viruses, especially for RV and HRSV, could greatly help to reduce the burden of wheezing in children. Preventive measures for HRSV should be directed to at risk populations, such as children ≤ 2 years. While molecular detection may illuminate which viruses are associated with pediatric wheezing, greater attention is needed to understand if and how these viruses cause wheezing in children.

## Supporting information

**S1 Fig. Global prevalence of rhinovirus in people with wheezing disorders.**
(PDF)

**S2 Fig. Global prevalence of human respiratory syncytial virus in people with wheezing disorders.**
(PDF)

**S3 Fig. Global prevalence of human bocavirus in people with wheezing disorders.**
(PDF)

**S4 Fig. Global prevalence of adenovirus in people with wheezing disorders.**
(PDF)

**S5 Fig. Global prevalence of influenza in people with wheezing disorders.**
(PDF)

**S6 Fig. Global prevalence of human metapneumovirus in people with wheezing disorders.**
(PDF)

**S7 Fig. Global prevalence of enterovirus in people with wheezing disorders.**
(PDF)

**S8 Fig. Global prevalence of human parainfluenzavirus in people with wheezing disorders.**
(PDF)

**S9 Fig. Global prevalence of human coronavirus in people with wheezing disorders.**
(PDF)

**S10 Fig. Funnel plot for publication for human respiratory syncytial virus in people with wheezing disorders.**
(PDF)

**S11 Fig. Funnel plot for publication for human metapneumovirus in people with wheezing disorders.**
(PDF)

**S12 Fig. Funnel plot for publication for influenza in people with wheezing disorders.**
(PDF)

**S13 Fig. Funnel plot for publication for rhinovirus in people with wheezing disorders.**
(PDF)

**S14 Fig. Funnel plot for publication for human adenovirus in people with wheezing disorders.**
(PDF)

**S15 Fig. Funnel plot for publication for human bocavirus in people with wheezing disorders.**
(PDF)

**S16 Fig. Funnel plot for publication for human parainfluenzavirus in people with wheezing disorders.**
(PDF)

**S17 Fig. Funnel plot for publication for human coronavirus in people with wheezing disorders.**
(PDF)

**S18 Fig. Funnel plot for publication for enterovirus in people with wheezing disorders.**
(PDF)

**S1 Table. Preferred reporting items for systematic reviews and meta-analyses checklist.**
(PDF)

**S2 Table. Search strategy in medline (Pubmed).**
(PDF)

**S3 Table. Items for risk of bias assessment.**
(PDF)

**S4 Table. Main reasons of exclusion of eligible studies.**
(PDF)

**S5 Table. Individual characteristics of included studies.**
(PDF)

**S6 Table. Subgroup prevalence of respiratory viral infections in people with wheezing.**
(PDF)

**S7 Table. Univariable and multivariable metaregression analysis on the prevalence of respiratory viruses in people with wheezing disorders.**
(PDF)

## Acknowledgments

We greatly appreciate the manuscript proofreading and excellent advice given by Dr Alexey Clara and Dr Karen A. Alroy.

## Author Contributions

**Conceptualization:** Cyprien Kengne–Nde, Sebastien Kenmoe, Richard Njouom.

**Data curation:** Cyprien Kengne–Nde, Sebastien Kenmoe.

**Formal analysis:** Cyprien Kengne–Nde, Sebastien Kenmoe.

**Investigation:** Cyprien Kengne–Nde, Sebastien Kenmoe.

**Methodology:** Cyprien Kengne–Nde, Sebastien Kenmoe, Abdou Fatawou Modiyinji, Richard Njouom.

**Project administration:** Cyprien Kengne–Nde, Sebastien Kenmoe, Richard Njouom.

**Supervision:** Cyprien Kengne–Nde, Sebastien Kenmoe, Richard Njouom.

**Validation:** Cyprien Kengne–Nde, Sebastien Kenmoe, Abdou Fatawou Modiyinji.

**Writing – original draft:** Sebastien Kenmoe.

**Writing – review & editing:** Cyprien Kengne–Nde, Sebastien Kenmoe, Abdou Fatawou Modiyinji, Richard Njouom.

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
