## [Decision Letter · Decision Letter 0]

22 Jul 2020

PONE-D-20-03613

Prevalence of respiratory viruses using molecular detection in children with wheezing, a systematic review and meta–analysis

PLOS ONE

Dear Dr. Njouom,

Thank you for submitting your manuscript to PLOS ONE. Please accept my apologies for the extended delay in getting a decision to you. 

Your manuscript was assessed by three external reviewers, whose reports are printed below (please see the attached Word document for reviewer 3's full report). After careful consideration of the reports, we feel your manuscript has merit but does not fully meet PLOS ONE’s publication criteria as it currently stands. Therefore, we invite you to submit a revised version of the manuscript that addresses the points raised during the review process.

I would encourage you to focus specifically on the following aspects, to ensure that your revised manuscript meets our publication criteria (as well as addressing all of the other reviewer comments):

- regarding the statistical analysis, please provide additional information on study heterogeneity and corrections for multiple comparisons, as requested by reviewer 3.

- please include some discussion of how the specific tests used in the different studies have affected your results: as reviewer 2 says, which tests were performed will inevitably influence which viruses are detected. Further discussion of the tests will also help to address reviewer 1's point on different molecular detection methods and how their sensitivity has changed over time.

We look forward to receiving your revised manuscript.

Kind regards,

Joseph Donlan

Academic Editor

PLOS ONE

Journal Requirements:

3. We note that this manuscript is a systematic review or meta-analysis; our author guidelines therefore require that you use PRISMA guidance to help improve reporting quality of this type of study. Please upload copies of the completed PRISMA checklist as Supporting Information with a file name “PRISMA checklist”.

Reviewers' comments:

Reviewer's Responses to Questions

**Comments to the Author**

1. Is the manuscript technically sound, and do the data support the conclusions?

Reviewer #1: Partly

Reviewer #2: Yes

Reviewer #3: Yes

2. Has the statistical analysis been performed appropriately and rigorously? 

Reviewer #1: Yes

Reviewer #2: Yes

Reviewer #3: Yes

3. Have the authors made all data underlying the findings in their manuscript fully available?

Reviewer #1: Yes

Reviewer #2: Yes

Reviewer #3: Yes

4. Is the manuscript presented in an intelligible fashion and written in standard English?

Reviewer #1: Yes

Reviewer #2: Yes

Reviewer #3: Yes

5. Review Comments to the Author

Reviewer #1: The authors performed an electronic in Pubmed and Global Index Medicus on 01 July 2019

and manual search of studies that have detected common respiratory

viruses in children ≤18 years with wheezing. only studies using molecular

detection methods, namely polymerase chain reaction (PCR) assays were included in this meta-analysis. The authors concluded that rhinovirus and respiratory syncytial virus may contribute to the etiology of wheezing in children. While the clinical implications of molecular detection of respiratory viruses remains an interesting question, this study helps to illuminate the potential of role respiratory viruses in pediatric wheezing.

This meta-analyses is quite comprehensive, and the result are quite interesting, They have tried to include non-English literature. This is good. But for the molecular detection method, they should describe more clearly how they defined it, not just said PCR methods. In addition, the participants was from 1992 to 2014. The sensitivity of molecular diagnostic techniques in recent years should be much higher than old days. The authors should discuss about this issue.

Reviewer #2: This is a well written systematic review that is easy to read. It is on an important topic but one that isn't really currently disputed - the role of respiratory viruses in causing acute wheezing episodes in children. The paper provides important new data synthesis on geographic area of study.

What's currently under debate is the role of bacterial infections in 'seemingly viral induced' wheeze events and bacterial/viral co-infection - but this is not what this paper is about.

Some suggestions that might improve the paper

1] It seems that the included studies used a range of different viral PCR tests (some with more than others). So - if a study only looked for RSV and RV these will be the only viruses that will be found - but the proportion with nil detected should be larger that eg in a study that looked at panel of N=17 respiratory viruses. It would be help if Table 1.5 (supplement) could have this data added. This Table could first be simplified by merging the study population 'Age' into one column (median, IQR) and the two columns Design/Timing could also be merged eg 'retrospective clinical trial'. The 2 freed up columns could then contain a list of viruses studied in each study and a second column with % with any virus detected, or % with no virus detected.

2]I think there needs to be some discussion about 'acute bronchiolitis' - this wasn't a search term and ideally shouldn't be BUT it does muddy the waters with respect to the <2 years data. I think to add bronchiolitis term - would bring up a host of new papers BUT would also be addressing a different question - so I think the likely overlap with acute bronchiolitis needs simply discussed. One way around this could also be to exclude studies with children wheezing less than 1 year - but it may not be possible to extract this data.

3] Given the project / paper title - if the Editors approved (from space in journal) I would like to see both Table 1.5 and Supplement Fig 1 in the main body rather than having to dig deeper into the Supplementary data to access this.

Very minor points

1] Abstract - Results: some data and text have no space between and I2 ('2' should be superscript)

2] Methods - Exclusion criteria: the last sentence needs improved for meaning ie that papers with multiple follow up times were only used once.

3] Table 1 needs simplified - do not need a separate % column eg Age 10y (30%) and these global percentage needs round to whole figure. Or Age (median 1.5 (IQR: 1.0 to 3.2) years.

Sample type ? whats the difference between an Oropharyngeal and a Throat swab? ?could these two be merged.

Reviewer #3: The main issue for major revision point to the calculations of prevalence values, which have been done several times. This raises the concern for multiple comparisons where corrections should be applied. Conclusions made then need to be modified.

6. PLOS authors have the option to publish the peer review history of their article (what does this mean?). If published, this will include your full peer review and any attached files.

Reviewer #1: No

Reviewer #2: No

Reviewer #3: No

---

## [Author Response · Author response to Decision Letter 0]

27 Aug 2020

Review Comments to the Author

Editor: I would encourage you to focus specifically on the following aspects, to ensure that your revised manuscript meets our publication criteria (as well as addressing all of the other reviewer comments):

- regarding the statistical analysis, please provide additional information on study heterogeneity and corrections for multiple comparisons, as requested by reviewer 3.

Authors: Regarding heterogeneity, we have now added univariate and multivariate metaregression to explain the sources of heterogeneity (S7 Table).

Regarding multiple comparisons, the statistical approach used incorporates a correction for this aspect. We have now presented the summary figure of all viruses in the main manuscript and changed the figure present in the supporting information to figures dealing with the viruses individually. The results of viruses treated individually or in groups are no different. This could suggest that there is no influence of the contribution of the same participant for more than one prevalence of the virus. Thank you.

- please include some discussion of how the specific tests used in the different studies have affected your results: as reviewer 2 says, which tests were performed will inevitably influence which viruses are detected. Further discussion of the tests will also help to address reviewer 1's point on different molecular detection methods and how their sensitivity has changed over time.

Authors: We considered the year of publication in the metaregression and there is no effect of this parameter on the prevalence of different viruses. This could suggest that the sensitivity of the molecular methods reported in the studies has not significantly changed for our contemporary data. In addition, only classical PCR and real-time PCR were used in the articles included in the review. Thank you.

Reviewer #1: The authors performed an electronic in Pubmed and Global Index Medicus on 01 July 2019 and manual search of studies that have detected common respiratory viruses in children ≤18 years with wheezing. only studies using molecular detection methods, namely polymerase chain reaction (PCR) assays were included in this meta-analysis. The authors concluded that rhinovirus and respiratory syncytial virus may contribute to the etiology of wheezing in children. While the clinical implications of molecular detection of respiratory viruses remains an interesting question, this study helps to illuminate the potential of role respiratory viruses in pediatric wheezing.

This meta-analyses is quite comprehensive, and the result are quite interesting, They have tried to include non-English literature. This is good. 

Authors: We thank the Reviewer for this summary and appreciation.

But for the molecular detection method, they should describe more clearly how they defined it, not just said PCR methods. 

Authors: We included studies using Polymerase Chain Reaction (PCR) only. PCR is actually the most widely used technique for diagnosing respiratory viruses. We made this clarification in the title and the abstract. We also added the detection assay variable used in the database of individual characteristics of the included studies (Supplementary table 5). Thank you.

In addition, the participants was from 1992 to 2014. The sensitivity of molecular diagnostic techniques in recent years should be much higher than old days. The authors should discuss about this issue.

Authors: We considered the year of publication in the metaregression and there is no effect of this parameter on the prevalence of different viruses. This could suggest that the sensitivity of the molecular methods reported in the studies has not significantly changed for our contemporary data. In addition, only classical PCR and real-time PCR were used in the articles included in the review. Thank you.

Reviewer #2: This is a well written systematic review that is easy to read. It is on an important topic but one that isn't really currently disputed - the role of respiratory viruses in causing acute wheezing episodes in children. The paper provides important new data synthesis on geographic area of study. What's currently under debate is the role of bacterial infections in 'seemingly viral induced' wheeze events and bacterial/viral co-infection - but this is not what this paper is about. 

Authors: We thank the Reviewer for this summary and appreciation. 

Some suggestions that might improve the paper

1] It seems that the included studies used a range of different viral PCR tests (some with more than others). So - if a study only looked for RSV and RV these will be the only viruses that will be found - but the proportion with nil detected should be larger that eg in a study that looked at panel of N=17 respiratory viruses. 

Authors: For each included study we collected data on the specific prevalence of individual viruses. This suggests that the number of viruses sought in the study might not influence the prevalence obtained. Thank you.

It would be help if Table 1.5 (supplement) could have this data added. This Table could first be simplified by merging the study population 'Age' into one column (median, IQR) and the two columns Design/Timing could also be merged eg 'retrospective clinical trial'. The 2 freed up columns could then contain a list of viruses studied in each study and a second column with % with any virus detected, or % with no virus detected.

Authors: Supplementary Table 5 has been modified in accordance with the suggestion, Thank you.

2]I think there needs to be some discussion about 'acute bronchiolitis' - this wasn't a search term and ideally shouldn't be BUT it does muddy the waters with respect to the <2 years data. I think to add bronchiolitis term - would bring up a host of new papers BUT would also be addressing a different question - so I think the likely overlap with acute bronchiolitis needs simply discussed. One way around this could also be to exclude studies with children wheezing less than 1 year - but it may not be possible to extract this data.

Authors: We added this sentence in the discussion to address this, thank you. 

“Wheezing is considered bronchiolitis in children <1 year of age [71]. Almost all of the studies included in this review have children <1 year of age that we cannot exclude and could therefore be the source of additional heterogeneity.”

3] Given the project / paper title - if the Editors approved (from space in journal) I would like to see both Table 1.5 and Supplement Fig 1 in the main body rather than having to dig deeper into the Supplementary data to access this.

Authors: Although the size of Supplementary figure 1 and Supplementary fable 5 are large, I have no opposition if the Editor allow insertion of these illustrations into the main manuscript, thank you.

Very minor points

1] Abstract - Results: some data and text have no space between and I2 ('2' should be superscript)

Authors: Thank you, we revised as requested.

2] Methods - Exclusion criteria: the last sentence needs improved for meaning ie that papers with multiple follow up times were only used once.

Authors: Thank you, we revised as suggested.

“Included articles with multiple follow up time were used only once for each virus.”

3] Table 1 needs simplified - do not need a separate % column eg Age 10y (30%) and these global percentage needs round to whole figure. Or Age (median 1.5 (IQR: 1.0 to 3.2) years.

Sample type? whats the difference between an Oropharyngeal and a Throat swab? ?could these two be merged.

Authors: Thank you, we revised as suggested.

Reviewer #3: The main issue for major revision point to the calculations of prevalence values, which have been done several times. This raises the concern for multiple comparisons where corrections should be applied. Conclusions made then need to be modified.

Authors: Regarding multiple comparisons, the statistical approach used incorporates a correction for this aspect. We have now presented the summary figure of all viruses in the main manuscript and changed the figure present in the supporting information to figures dealing with the viruses individually. The results of viruses treated individually or in groups are no different. This could suggest that there is no influence of the contribution of the same participant for more than one prevalence of the virus. Thank you.

LINE numbers would have facilitated speed of the review.

Authors: We added the line numbers, thank you.

Spacings and superscript issues dot this manuscript. Please fix.

Authors: We removed the spaces and applied the superscript where necessary, thank you.

Figure 2. 

Is there a way to simplify the items in Figure 2? It looks like a hybrid between a forest plot and table and would the authors express the significance in terms of the p-value? If random-effects were used throughout not necessary to repeat the term

Authors: We thank the reviewer for this comment but despite all the investigations we have made, we are unable to modify the figure that is produced by the software. Thank you.

It seems that heterogeneity is substantial did the authors take steps to adjust for heterogeneity? Were sources of heterogeneity identified?

Authors: Regarding heterogeneity, we have now added univariate and multivariate metaregression to explain the sources of heterogeneity (S7 Table). Thank you.

PAGE 8

The studies were published between 2002 and 2018. Most were cross–sectional studies (16; 48.5%),

“Most” is followed by “48.5%”. This is under 50%, how could it be most? Did I miss something here? 

Authors: Categories with percentages <50% have been removed from the description of major groups, thank you.

Table 1. Sociodemographic and clinical characteristics of included studies

Age in years is expressed in median IQR, why not mean SD? Did the authors perform test for normal distribution, if so, please show the results.

Authors: We added the normality test result in the manuscript, thank you.

“According to the Shapiro-Wilk test, the age distribution did not follow a normal distribution (p <0.001) and therefore we expressed the age as the median and interquartile range.”

Table 2. Overall meta-analysis for global prevalence of respiratory viral infections in children with wheezing 

1.6. Supplemental table 6. Subgroup prevalence of respiratory viral infections in people with wheezing

The authors generated several prevalence values with indications of their significance or its absence. Did the authors perform corrections for multiple comparisons?

Authors: Regarding multiple comparisons, the statistical approach used incorporates a correction for this aspect. We have now presented the summary figure of all viruses in the main manuscript and changed the figure present in the supporting information to figures dealing with the viruses individually. The results of viruses treated individually or in groups are no different. This could suggest that there is no influence of the contribution of the same participant for more than one prevalence of the virus. Thank you.

---

## [Decision Letter · Decision Letter 1]

26 Nov 2020

Prevalence of respiratory viruses using polymerase chain reaction in children with wheezing, a systematic review and meta–analysis

PONE-D-20-03613R1

Dear Dr. Njouom,

We’re pleased to inform you that your manuscript has been judged scientifically suitable for publication and will be formally accepted for publication once it meets all outstanding technical requirements.

Kind regards,

Michael D Shields, MD

Guest Editor

PLOS ONE

Additional Editor Comments (optional):

Thank you for the response to the peer reviewers.

It would be better if the Tables S6/7 had the 'p-values' not highlighted in bold (unless the bold is indicating that the value is statistically significant ie < 0.05).

I'm now happy to recommend to the Editor that this paper should be considered for publication

Reviewers' comments:

Reviewer's Responses to Questions

**Comments to the Author**

1. If the authors have adequately addressed your comments raised in a previous round of review and you feel that this manuscript is now acceptable for publication, you may indicate that here to bypass the “Comments to the Author” section, enter your conflict of interest statement in the “Confidential to Editor” section, and submit your "Accept" recommendation.

Reviewer #3: All comments have been addressed

2. Is the manuscript technically sound, and do the data support the conclusions?

Reviewer #3: Yes

3. Has the statistical analysis been performed appropriately and rigorously? 

Reviewer #3: Yes

4. Have the authors made all data underlying the findings in their manuscript fully available?

Reviewer #3: No

5. Is the manuscript presented in an intelligible fashion and written in standard English?

Reviewer #3: Yes

6. Review Comments to the Author

Reviewer #3: The authors have responded to my comments regarding the methodology. Could be better, since they said about being unable to revise Figure 2.

Kindly supply S7 Table. Need to examine this table.

7. PLOS authors have the option to publish the peer review history of their article (what does this mean?). If published, this will include your full peer review and any attached files.

Reviewer #3: No

---

## [Editor Report · Acceptance letter]

3 Dec 2020

PONE-D-20-03613R1 

Prevalence of respiratory viruses using polymerase chain reaction in children with wheezing, a systematic review and meta–analysis  

Dear Dr. Njouom:

I'm pleased to inform you that your manuscript has been deemed suitable for publication in PLOS ONE. Congratulations! Your manuscript is now with our production department. 

Kind regards, 

on behalf of

Professor Michael D Shields 

Guest Editor

PLOS ONE